

# A Dynamic Hierarchical Bayesian Approach for Forecasting Vegetation Condition

Edward E. Salakpi[1], Peter D. Hurley[1,2], James M. Muthoka[3], Andrew Bowell[1,2], Seb Oliver[1,2], and Pedram Rowhani[3]

[1]The Data Intensive Science Centre, Department of Physics and Astronomy, University of Sussex, Brighton BN1 9QH, UK
[2]Astronomy Centre, Department of Physics and Astronomy, University of Sussex, Brighton BN1 9QH, UK
[3]Department of Geography, School of Global Studies, University of Sussex, Brighton BN1 9QJ, UK

**Correspondence:** Edward E. Salakpi (e.salakpi@sussex.ac.uk)

**Abstract.** Agricultural drought, which occurs due to a significant reduction in the moisture required for vegetation growth, is the most complex amongst all drought categories. The onset of agriculture drought is slow and can occur over vast areas with varying spatial effects, differing in areas with a particular vegetation land cover or specific agro-ecological sub-regions. These spatial variations imply that monitoring and forecasting agricultural drought require complex models that consider the

spatial variations in a given region of interest. Hierarchical Bayesian Models are suited for modelling such complex systems. Using partially pooled data with sub-groups that characterise spatial differences, these models can capture the sub-group variation while allowing flexibility and information sharing between these sub-groups. This paper's objective was to improve the accuracy and precision of agricultural drought forecast in spatially diverse regions with a Hierarchical Bayesian Model. Results showed that the Hierarchical Bayesian Model was better at capturing the variability for the different agro-ecological

zones and vegetation land covers compared to a regular Bayesian Auto-Regression Distributed Lags model. The forecasted vegetation condition and associated drought probabilities were more accurate and precise with the Hierarchical Bayesian Model at 4 to 10 weeks lead times. Forecasts from the hierarchical model exhibited higher hit rates with a low probability of false alarms for drought events in semi-arid and arid zones. The Hierarchical Bayesian Model also showed good transferable forecast skills over counties not included in the training data.

## 1  Introduction

Drought is a naturally occurring phenomenon that affects the food security of approximately 55 million people annually and can severely impact a country's economy (Deleersnyder, 2018; Nicolai-Shaw et al., 2017). Drought, in most cases, is associated with below-average precipitation and is referred to as meteorological drought. Prolonged meteorological drought event mainly leads to a significant reduction in the amount of soil moisture required for vegetation growth, thus resulting

in an agricultural drought (Heim, 2002; Boken et al., 2005). Hence, agricultural drought events are considered a physical manifestation of meteorological drought (Boken et al., 2005). Agricultural drought, which is the focus of this paper, is the most complex amongst the drought categories (Boken et al., 2005). Its onset can be slow and can occur in vast areas with varying spatial impact (Boken et al., 2005). For instance, the impact of drought may differ within a given region depending on





whether they are dominated by trees, grasslands or croplands. Spatial differences in drought impact can also arise due to the
varied agro-ecological sub-regions within an affected area. These differences indicate that Early Warning Systems (EWS) for
agricultural drought will require very complex models.

Drought EWS have been recognised by global initiatives like the United Nations Sustainable Development Goals (SDG) for
effective drought monitoring and hazard preparedness (IISD, 2018). As such, international agencies like United Nations De-
velopment Programme (UNDP) and the United States Agency for International Development (USAID) [1] mandated to monitor
drought hazards have developed and deployed several EWS. These systems assist drought management officials and people
living in drought-prone communities to prepare for hazardous events (UN, 2018). The Famine Early Warning Systems Net-
work (FEWS NET) [2] is an example of such EWS. The system, developed by the USAID, utilises household data together
with agro-climatic indicators and vegetation health to monitor drought and its impact (FEWSNET, 2021). However, drought
forecast for anticipatory action via the FEWS NET platform is mainly based on expert judgement (Funk et al., 2019) rather
than the use of advanced statistical methods or machine learning models.

Recent advances in computational power and processing hardware have enabled researchers to develop and deploy machine
learning models (Bishop, 2006) such as Support Vector Machines (Shao and Lunetta, 2012) and various neural network archi-
tectures (Da Silva et al., 2017). Machine learning models enables the construction of predictive or prescriptive models using
advanced statistical methods to capture hidden patterns in data (Bishop, 2006). In the field of drought research, most of the data
used within machine learning models come from satellite Earth Observation (EO) images. These datasets are available over
long temporal periods, cover vast areas and are easy to access. Therefore, they provide a cost-effective way of developing mod-
els for monitoring and forecasting drought events over vast regions. Examples of such EO datasets include precipitation, soil
moisture levels, Normalised Difference Vegetation Index (NDVI), Enhanced Vegetation Index (EVI) and Vegetation Condition
Index (VCI) (Kogan, 1995) all derived from remotely sensed EO data. Nay et al. (2018), for instance, used Gradient Boosting
Machine to forecast EVI with lagged spectral bands from the Moderate Resolution Imaging Spectroradiometer (MODIS) EO
data. Tian et al. (2019) worked on forecasting dryland vegetation condition using NDVI via an Eco-hydrological model driven
by surface water extent also derived from MODIS images. Others include Barrett et al. (2020) and Adede et al. (2019) who
applied Gaussian Processes and Artificial Neural Networks respectively in their research to develop robust models for short to
medium-term forecasts of vegetation conditions. All the models used in the cited works were mainly implemented by aggre-
gating data over similar land cover types and Agro-Ecological Zones (AEZ). The differences in the AEZs or land covers within
the region were not considered.

In a previous study (Salakpi et al., 2021), we used a Bayesian regression method to model the relationship between biophys-
ical drivers and their effect on forecasting vegetation conditions. The approach was based on the classical 'No-pooling' method
(See figure 1), where we fitted separate regression models to data extracted from their respective regions. Pixels representing
the biophysical indicators and vegetation conditions were sampled for different land cover and aggregated over the regions
of interest. The modelling approach also treated the effects of climate and other biophysical factors on vegetation conditions

---

[1]https://usaid.gov/
[2]https://fews.net/




independently for each region. The models were very skilful for medium to long term forecasts, but forecasts over regions with extensive cloud cover suffered due to the lack of data.

Although known to vary over the different regions, the effects of biophysical indicators on vegetation also show some
similarities across the different regions (Vicente-Serrano, 2007). Data for such analysis can be pooled over all the regions of interest and analysed via the 'Complete-pooling' modelling approach to capture these similarities. This approach allows information sharing between the regions of interest, which is an advantage over the 'No-pooling' approach (Gelman and Hill, 2006). However, the 'Complete-pooling' method is not very useful when the pooled data has sub-groupings, e.g., a pooled soil moisture data from different regions with varied land cover types. In such a case, a more advanced approach would be
to combine the strengths of both the 'No-pooling' and 'Complete-pooling' methods into a model known as a 'Partial-pooled' model or 'Hierarchical model' (Gelman and Hill, 2006; Gelman et al., 2013). The hierarchical approach, which we demonstrate in this paper, enables flexibility between the sub-groups while treating them independently at the same time (Gelman and Hill, 2006). A Hierarchical Model, when implemented within a Bayesian framework, is referred to as a Hierarchical Bayesian Model (HBM) (Gelman et al., 2013). HBMs have in recent times been recognised as a powerful approach for modelling and
analysing very complex data. They have been extensively used for research in fields like Astrophysics, Neuroscience and Genetics (Sánchez and Bernstein, 2019; George and Hawkins, 2005; Storz and Beaumont, 2002). Although not commonly used in the study of vegetation dynamics and drought monitoring, Senf et al. (2017) used an HBM to study the spatial and temporal variation in broad-leaved forests phenology using Landsat data.

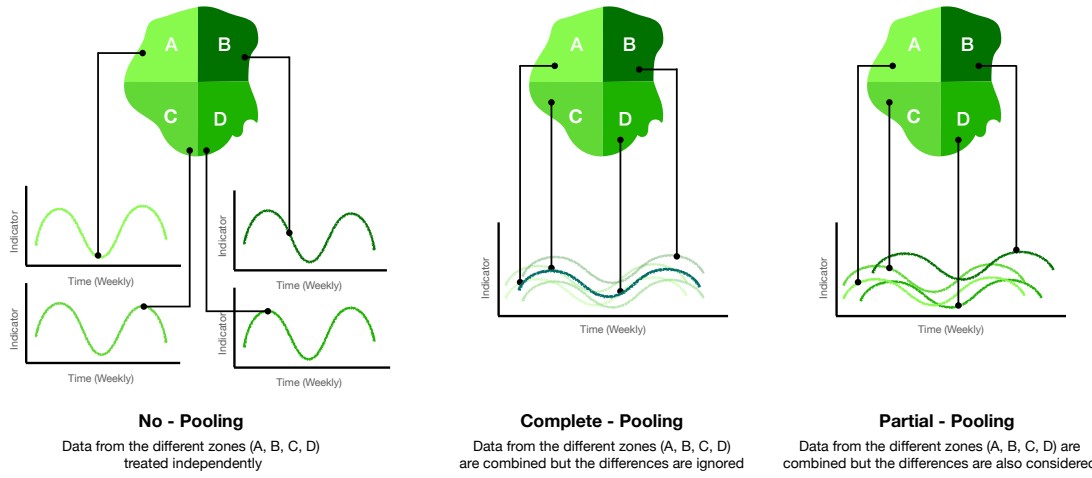

**Figure 1.** Figure illustrating the concept of 'No-Pooling', 'Complete-Pooling' and 'Partial-Pooling' of the data.


The HBM is an extension of the regular Bayesian regression where model parameters differ based on the variations within
a given dataset (Gelman et al., 2013; Gelman and Hill, 2006). Thus, this paper sought to test the concept of forecasting
VCI, an EO based agricultural drought indicator, with an HBM and answer the following question. *'Can we improve forecast
accuracy and precision by separately learning parameters for the effects of lagged precipitation and soil moisture on vegetation
conditions in each AEZ or over varied land cover types?'.*

Another advantage of using the HBM is its transferability (Senf et al., 2017). Transfer learning in this context refers to the
process where models trained on a given dataset can be re-used to make predictions on different but related data that was not
part of the training set (Yang et al., 2017b). The partially pooled data used in HBMs makes it suitable for transfer learning
primarily because the training data are pooled from multiple regions, and the sub-groupings within the data are the same for
the non-training sample data (Rosenstein et al., 2005).

Our objectives for this proof-of-concept are to:

– improve the forecast accuracy and precision of Bayesian Auto-Regression Distributed Lags (BARDL) model with a
Hierarchical Bayesian Model in regions with diverse AEZs, and land covers.

– test the transfer learning property of hierarchical model that enables pre-trained models to be used on similar data from
a different location without the need to retrain the model (Yang et al., 2017a).

## 2 Study Area and Data

### 2.0.1 Study Area

To test our concept of forecasting vegetation condition with HBM, we sampled data from some selected counties in Kenya
(Baringo, Kitui, Marsabit, Narok, Tana-River, Turkana), shown in figure 2 with red boundary lines. The selected counties have
diverse land use and land covers (LULC), ranging from crops to evergreen forests. These counties also have varied AEZs with
rainfall and temperature patterns ranging from moderate to extreme. During the short and long rainfall seasons, annual mean
precipitation range from 20mm to 200mm. Temperature across these counties also range from as low as $10^oC$ to $40^oC$ (Ayugi
et al., 2016). The main economic activity in these counties is agriculture, predominantly agro-pastoral practices (Gebremeskel
et al., 2019; Vatter, 2019). However, extreme climatic variations make this region prone to prolonged drought events, and the
impact of these dry spells vary over the various land covers within the AEZs.

We selected only six counties because the algorithm used for parameter sampling by the HBM can be very time-consuming
when the input data is more than 10,000 records. The sampling time is also mainly due to the complex nature HBMs.


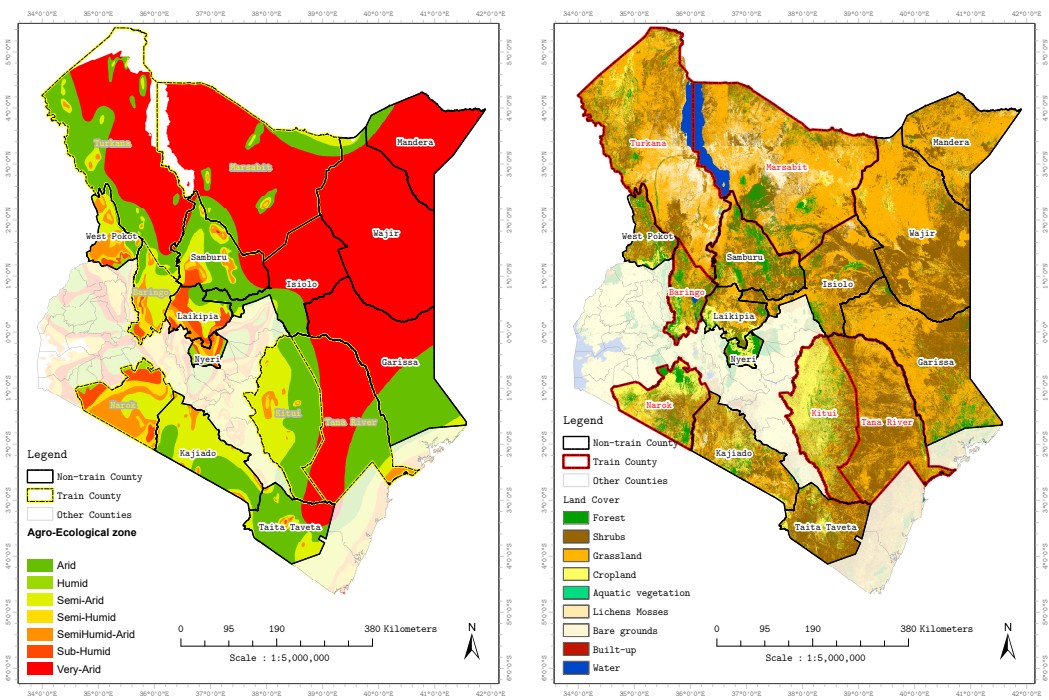

**Figure 2.** Maps of Kenya showing Agro-Ecological Zones (AEZ) and Land Cover maps for the counting from which pixels were sampled. Kenya AEZ boundary maps credit: IGAD Climate Prediction and Application Centre (ICPAC). Land Cover map credit: European Space Agency (ESA), Climate Change Initiative (CCI)

## 2.1 Data

Aside the AEZ boundary shapefiles, all the dataset used in this proof-of-concept is the same data used and described in a previous study here: (Salakpi et al., 2021).

## 2.2 Agro-Ecological Zones & Vegetation Land Covers

Two HBMs were developed in this study, one based on AEZs and the other on land covers. AEZs are geographical areas characterised by similar climatic patterns and soil moisture levels suitable for agriculture and vegetation growth. These zones were created by the Food and Agriculture Organization (FAO) in collaboration with International Institute for Applied Systems Analysis (IIASA) and are based on a framework that utilises a series of models with climate and land use information to quantify and map out the regions (Fischer et al., 2000). The zones are categorised as Humid, Semi-Humid, Arid, Semi-Arid

and Very Arid. These AEZs, from their definition, exhibit distinct climate properties; thus, a modelling approach that can separately learn parameters for the effects of precipitation and soil moisture on vegetation conditions based on the difference AEZs can give a more accurate VCI forecast.


The AEZs in our study area include:

**Table 1.** Table describing the Agro-Ecological Zone, vegetation type and annual rainfall levels.

| Zone Classification | Vegetation Type | Average Annual Zone Rainfall (mm) |
|---|---|---|
| Humid | Moist Forest | 1100-2700 |
| Sub-Humid | Moist and Dry Forest | 1000-1600 |
| Semi-Humid | Dry Forest and Moist Woodlands | 800-1400 |
| Semi-Humid to Arid | Dry Woodland and Bush lands | 600-1100 |
| Arid | Bush, Grass and Shrublands | 450-900 |
| Semi-Arid | Bush, Grass and Shrublands | 300-500 |
| Very-Arid | Desert, Sparse grass and shrub | 150-350 |

Source: Sombroek et al. (1982)

Most drought-prone ROIs are made of diverse vegetation covers; these include Tree Covers (Forests), Grasslands, Shrubs and Croplands. The impact of drought on these land cover types varies both spatially and temporally. Thus, a drought forecast model should consider the varying effects of the biophysical factors on the various land covers. Using an HBM framework in this context enables us to achieve this. Data corresponding to the various vegetation land covers was extracted with the Sentinel 2, 2016, Land Use and Land Cover (LULC) map [3].

## 3    Methodology

### 3.1    Data Pre-Processing

A major challenge with using optical EO images is cloud cover and cloud shadows. In addition, pixel reflectance values sometimes fall outside the meaningful range due to errors during the atmospheric and radiometric correction process. These clouded and poor-quality pixels were filtered out with the quality assurance maps that come with the images. Weekly averages of VCI, precipitation and soil moisture data corresponding to the vegetation land covers of interest were extracted from the selected counties using the European Space Agency (ESA) 2016 Sentinel 2 Land Use and Land Cover (LULC) map. Same data within the various AEZs were also extracted using the AEZ shapefiles produced by IGAD Climate Prediction and Application Centre (ICPAC) [4]. The temporal gaps, left by the removal of clouded pixels, were filled using the Radial Basis Function (BBF) interpolation method, which ensures values obtained through interpolation over wide intervals do not go beyond the valid ranges (Rippa, 1999). The noise resulting from optical instrument failures and gap-filling processes were reduced with a penalised least-squares method (Whittaker smoother) (Eilers, 2003; Klisch and Atzberger, 2016). A three-month (12 weeks)

---

[3]Visit this link (http://2016africalandcover20m.esrin.esa.int/) to learn more about the European Space Agency (ESA), Climate Change Initiative (CCI) Sentinel 2 Land Cover Map

[4]http://geoportal.icpac.net/layers/geonode%3Aken_aczones



rolling average was applied to the VCI to make it VCI3M primarily because our stakeholders use it for their EWS. Applying the rolling averages enhanced the persistence in the VCI. Three-month Precipitation (P3M) and Soil moisture (SM3M) were also computed for consistency. Finally, to avoid the influence of strong seasonal cycles on the forecast values and make data stationary, the VCI3M, P3M and SM3M data were converted to anomalies by subtracting their seasonal means before fitting to the HBM. After forecasting, the subtracted seasonal means for the VCI3M (for each AEZ and land cover) were added back. All the variables were also standardised by subtracting the mean and divided by the standard deviation to make the variable unitless and avoid the dominance of certain variables over others.

## 3.2 Forecast Model

The HBM implemented in this work was done via an Auto-Regressive Distributed Lag (ARDL) model (Gujarati, 2003). The ARDL*(p,q)* is a time series regression method used for multivariate time series analysis where the variable of interest (dependent variable) is modelled with its lags and that of additional explanatory variables (independent variable) (Gujarati, 2003). The *p* represents the number of lags for the independent variable used in the model, and the *q* is the auto-regressive part of the model, representing the lags of the dependent variable passed to the ARDL model. Within the HBM framework, a Bayesian probabilistic approach is used to infer model parameters instead of the Maximum likelihood approach. The data $Y$ for the model is partially pooled as $Y_{ij}$ where $i$ is the index of the variable (e.g. precipitation), and $j$ are the indices of the sub-groups (e.g. AEZs) within the data. This data structure enables parameter inference at both the global $\theta_i$ and sub-group levels $\theta_j$ at the same time as shown in figure 3. Using the Bayesian framework also allows us to incorporate informative priors into the parameter estimation process. Furthermore, we obtain a full posterior probability distribution for both the parameters and forecast values, instead of just point estimates, which enables gives a straightforward way to quantify forecast uncertainties (R. Ravines et al., 2006; Asaad and Magadia, 2019).





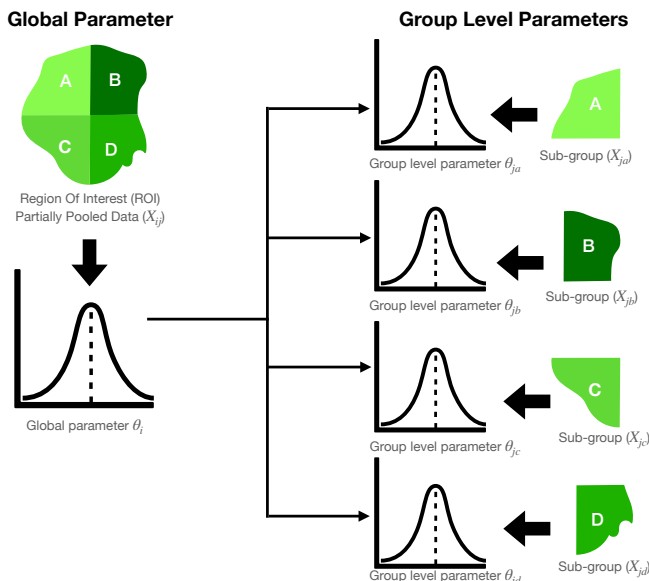

**Figure 3.** An illustration of the parameter structure of the Hierarchical Bayesian model based on partially pooled data ($Y_{ij}$). The global parameter ($\theta_i$) represents the average posterior parameter distribution over an entire region of interest, while the group level parameters $\theta_{j(abcd)}$ are the individual posterior parameter distributions inferred from the sub group data ($Y_{jabc}$) within the region of interest.

The Bayesian framework used for the parameter inference is based on Bayes' theorem in equation 1:

$$P(\theta|X_t) = \frac{P(X_t|\theta).P(\theta)}{P(X_t)} \tag{1}$$

where $X_t$ represents the input data of the ARDL model, $P(\theta|X_t)$) represents the probability of our model parameters given
our data $X_t$ also known as the posterior, $P(X_t|\theta)$ represents the probability of the data given the parameters referred to as the likelihood and $P(\theta)$ represents the prior belief about the parameters. $P(X_t)$ is the probability of data or evidence. The evidence is a normalisation term and usually ignored, making the posterior proportional to the likelihood and prior as seen in equation 2 (Lambert, 2018; McElreath, 2016).

$$P(\theta|X_t) \propto P(X_t|\theta).P(\theta) \tag{2}$$

It is important to note that working with the Bayes' framework allows us to explicitly define our prior beliefs about model parameters. These priors are then updated with the likelihood function to generate the posterior probability distribution when informed by observed data.

The HBM will enable us to fit the ARDL model by simultaneously inferring global parameters (Nodes A and B in figure 4) across the partially-pooled data as well as their sub-group level variations (Node G in figure 4) (Gelman and Hill, 2006). The


sub-group levels, in this case, refers to the different LULC or AEZs within our data. The varying effect of the sub-groups was incorporated into our HBM as categorical variables (Node K in figure 4).

      The HBM was based on an ARDL*(p=6,q=6)*, where the lagged of P3M, SM3M and VCI3M were all set to lags of 6 weeks. The nature of the input variables suggests a high likelihood for our model parameters to have a strong correlation. We addressed this by modelling our group-level parameters as a multivariate normal distribution using a Cholesky matrix decomposition as

hyper-priors (prior of a prior distribution) (Nodes C, D and E in figure 4) (McElreath, 2018). The Cholesky factorisation was used to transform the multivariate distribution to increase the efficiency of parameter sampling (Stan Development Team, 2018). However, since the HBM group-level parameters are modelled as conditional probabilities of the global parameters, the group level parameter tends not to separate well from the global mean. When this happens, the model does not converge, resulting in less precise forecasts. We handled this by introducing an offset factor (Node F in figure 4) to make the model non-centred

(Betancourt and Girolami, 2013). The global parameters were set to follow a normal distribution to enable parameter values to take on positive and negative values. Due to the hierarchical structure of the model parameters, global prior distribution usually serves as hyper-priors for the group-level parameters.

      Parameter approximation for the HBM was sampled with Hamiltonian Monte Carlo (HMC) algorithm (Hoffman and Gelman, 2014), an improved version of the classic Markov Chain Monte Carlo (MCMC) based on the notion of Hamiltonian

dynamic. For this research, however, the No-U-Turn Sampler (NUTS) (Hoffman and Gelman, 2014) version of HMC was used.

      Below (figure 4) is a Directed Acyclic Graph (DAG) schematic representation of an example of the HBM used for this study. From the HBM Directed Acyclic Graph (DAG) in figure 4:

- Node A is the global (Mean) regression intercept or ($\alpha_i$) parameter assumed to be Gaussian;

- Node B global (Mean) regression coefficients for each of the lagged input variables (precipitation and soil moisture) or ($\beta_i$) parameters for the 18 lagged variables (6 lags each for VCI3M, P3M, SM3M);

- Node C represent Cholesky covariance matrix used as hyper-priors for the group level $\alpha_j$ and $\beta_j$ parameters ;

- Nodes D and E are the Cholesky standard deviation and correlation from the matrix decomposition, respectively;

- Node F represent offset distribution (Gaussian) hyper-prior to make the model non-centred;

- Node G is the prior group-level parameters for $\alpha_j$ and $\beta_j$ parameters for each vegetation AEZ within our selected counties (i.e. Five AEZs ($\beta_j$) within each of the 18 ($\beta_i$) parameters plus 1 ($\alpha_i$));

- Node H represents the error term in the HBM regression;

- Node I is the likelihood function (equation 5) of the HBM regression and is based on ARDL*(p=6,q=6)* shown in equations 3 & 4;

- Node J is our lagged inputs datasets;





- – Node K is the categorical values that maps the input data to their respective AEZs;

- – Node L is the observed VCI3M values at an $i$ lead time.

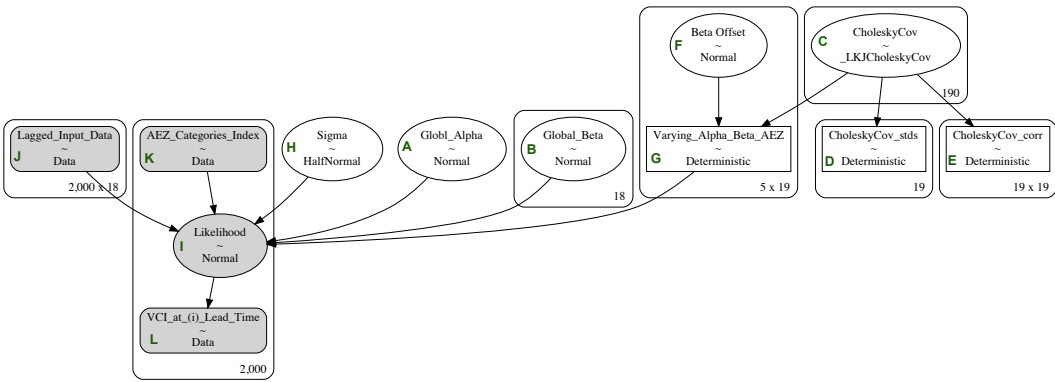

**Figure 4.** A Directed Acyclic Graph (DAG) schema representing the Hierarchical model based on varying Agro-Ecological Zones.

The Hierarchical BARDL model in this study was defined as:

$$D_{t+n} = \alpha_{j[i]} + \sum_{i=0}^{q} \beta_{j[d]} D_{t-q} + \sum_{i=0}^{p} \theta_{j[p]} P_{t-p} + \sum_{i=0}^{p} \delta_{j[s]} S_{t-p} + \epsilon_{t-p} \tag{3}$$

where $D_{t+n}$ is the VCI3M at $n$ weeks ahead, $D_{t-q}$ represent the data for lags 0 to $q$ of VCI3M (Dependent variable). $P_{t-p}$, $S_{t-p}$ are the lags 0, to $p$, P3M, and SM3M respectively. $\alpha_{j[i]}$ are the global ($i$) and group level ($j$) regression intercept, $\beta_{j[d]}$, $\theta_{j[p]}$, and $\delta_{j[s]}$ are the regression coefficients for the lagged P3M, and SM3M input variables at the global ($i$) and group level ($j$). $\epsilon_{t-p}$ is the regression error term. Equation (3) can be simplified as 4 and re-written as a Bayesian likelihood function $P(X_t|\theta)$ in equation 5:

$$D_{t+n} = \alpha_{j[i]} + \sum_{i=0}^{i} \beta_{j[i]} X_{t-i} + \epsilon_{t-i} \tag{4}$$

where $n$ is the lead time, $\beta_{j[i]}$ are the global and group level model parameters and $X_{t-i}$ represent the lagged input variables in equation 3.

$$P(X_t|\alpha_{j[i]}, \beta_{j[i]}, \sigma) \sim N(\alpha_{j[i]} + \sum_{i=0}^{i} \beta_{j[i]} X_{t-i}, \sigma_{t-i}) \tag{5}$$

were $\alpha_{j[i]} \sim N(\mu_{\alpha_i}, \sigma_{\alpha_i}^2), \beta_{j[i]} \sim N\mu_{\beta_i}, \sigma_{\beta_i}^2)$

and

$\sigma_{t-i} \sim HalfN(0, \sigma_d^2)$.





### 3.3 Forecasting and Model Evaluation

The forecast method used in this work was the direct multi-step forecast approach as described by Ben Taieb et al. (2010) and Ben Taieb and Hyndman (2014).

$$D_{t+n} = \sum_{i=0}^{i} \nu_i X_{t-i} + \epsilon_{t-i} \tag{6}$$

where $\nu_i$ are the model parameters and $X_{t-i}$ are the lagged inputs.

With this approach, separate models are fitted for every $n$ lead time forecast. Meaning, for each $n$ step forecast ahead ($D_{t-n}$), the observed VCI3M for the training dataset is shifted by $n$ weeks ahead from lag0 $X_{t-0}$ for all input variables.

After the parameter estimation via HMC sampling, the held-out dataset is passed to the fitted model (without the target variable) to produce forecast values for $n$ weeks ahead. The held out observed values and mean values of our forecast distributions were used to compute the coefficient of determination ($R^2$) (Equation 8) and Root Mean Squared Error (RMSE), (Equation 7) metrics for forecast evaluation. The $R^2$ score quantifies the variation in the observed data that the model could explain, while the RMSE measures the average difference between the observed and forecast values.

$$RMSE = \sqrt{\frac{\sum_{i=n}^{n}(y_i - f_i)^2}{n}} \tag{7}$$

where the $y_i$ are the observed data, $f_i$ are the forecasts and $n$ the total number of data points.

$$R^2 = 1 - \frac{\sum_i (y_i - f_i)^2}{\sum_i (y_i - \bar{y})^2}, \tag{8}$$

where the $y_i$ are the observed data, and the $f_i$ are the forecasts.

The forecast uncertainties were analysed with the Mean Prediction Interval Width (MPIW) and the Prediction Interval Coverage Probability (PICP) (Pang et al., 2018). The PICP computes the percentage of time the observed variable falls within a chosen prediction interval. The MPIW measures the mean distance between the upper (u) and lower (l) bound for a chosen prediction interval.

The MPIW was derived as follows:

$$MPIW_{t+n} = \frac{1}{N} \sum_{i=1}^{n} |u(D_i) - l(D_i)| \tag{9}$$

where $u(D_i)$ and $l(D_i)$ are the absolute upper and lower bounds values of the forecast distribution.

The PICP was derived as follows:

$$PICP_{t+n} = \frac{1}{N} \sum_{i=1}^{n} c_i \tag{10}$$





where $N$ is the number of forecast samples, $c_i$ is either 0 if the observed drought indicator at $D_{t+n}$ value falls outside the prediction interval, and $c_i$ is 1 if the observed value at is within the upper and lower bound of the forecast distribution.

Other forecast verification metrics used in this paper are the Receiver Operating Characteristic (ROC) curve (Wilks, 2006) curve and forecast probability Reliability Diagrams and Sharpness plots (Wilks, 2006; Jolliffe and Stephenson, 2012).

The ROC curve tells us the likelihood of a forecast being true (True Positive Rate (TPR)) for the given drought threshold and the probability of the forecast event being false (False Alarm Rate (FAR)). In addition, the Area Under the Curve (AUC) was also computed to determine the propensity of our model to separate drought events for the set threshold (Bradley, 1997).

The Reliability Diagram allows us to assess the accuracy of the forecast probability predicted by our model. The probability of a drought event is computed using the full posterior distribution of our forecasts at a given drought threshold. The same threshold is used to convert observed data into binary events where 0 indicates a 'No Drought' and 1 indicate a 'Drought' event. The forecast probabilities and observed binaries are binned into probability intervals and used to plot the forecast reliability diagrams. The reliability of the forecast is assessed by the number of times an observed event agrees with a given forecast probability (Wilks, 2006). The sharpness plots, on the other hand, tells the frequency with which a drought event is predicted within a probability bin (WWRP, 2009).

## 4 Results

Our dynamic HBM for forecasting VCI3M were tested on datasets based on their AEZs and vegetation land covers. Two models were developed, A BARDL model based on a 'No-pooling' approach as a base model and an HBM based on the 'Partial-pooling' approach. The BARDL model was used to forecast VCI3M for the different AEZs, referred to as 'BARDL-AEZ', and different land covers, referred to as 'BARDL-LC'. The HBM, which was modelled with partially pooled AEZ data, is referred to as 'HBM-AEZ' and the model-based partially pooled land covers data will be referred to as 'HBM-LC'. The results shown in this section are a comparison of BARDL-AEZ to HBM-AEZ and BARDL-LC to HBM-LC.

The aim of modelling with HBM was to capture information on the variations within our partially pooled AEZ and land cover data. Figure 5 shows the time series plots of the observed and forecasted VCI3M from the BARDL and HBM at 4 weeks lead time for the different AEZs. From the plots, it is clear that the temporal variation differs for the various AEZs. The forecast values from the HBM-AEZ (figure 5 to the right) were more accurate than the BARDL-AEZ.

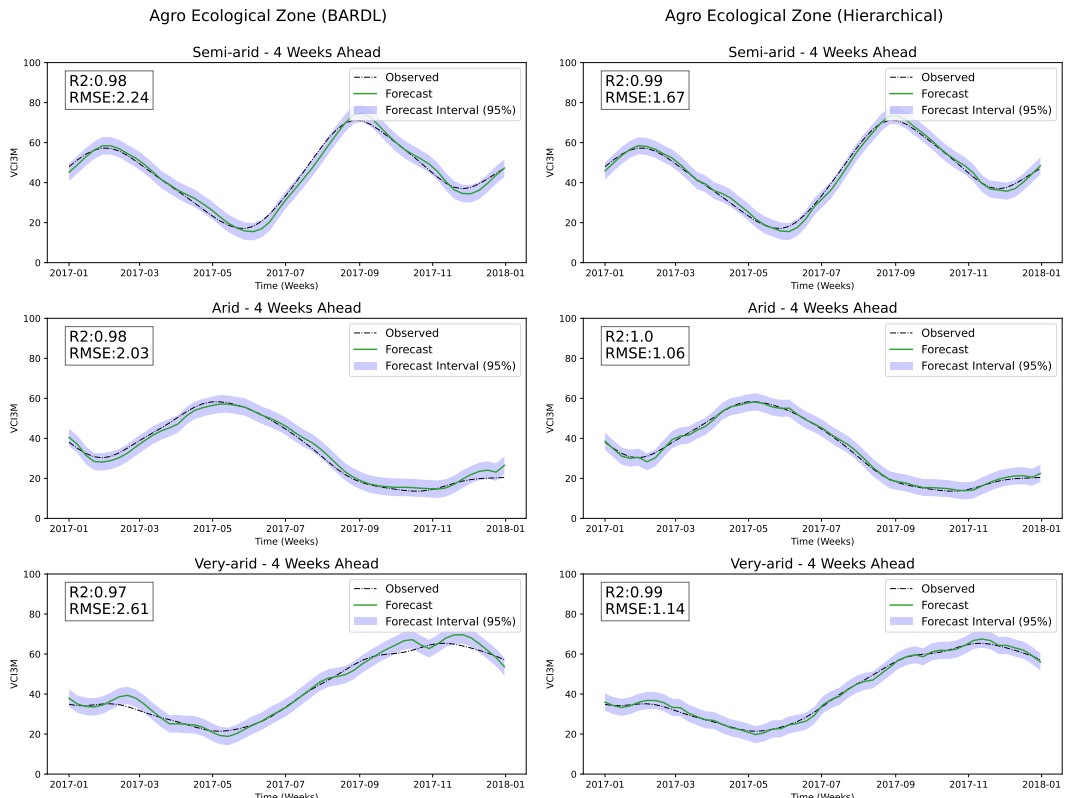

**Figure 5.** Time series Plots showing observed and forecast VCI3M at 4 week ahead in the semi-arid, arid and very-arid zones for the BARDL model (*left*) and HBM (*right*). The $R^2$ and RMSE metrics show that forecasts by the HBM are more accurate and have lower errors.

## 4.1 Model Performance for AEZ Based Models

The AEZ based models were used to forecast VCI3M for the Humid, Semi-Humid, Semi-Arid, Arid and Very-Arid zones. The $R^2$ scores and RMSE showed in figure 6 is for the Semi-Arid, Arid and Very-Arid zones since they were of most interest. The

results for humid zones can be seen in figure A1. Both $R^2$ scores and RMSE in figure 6 (A & B) showed that the HBM-AEZ model performed better than the BARDL-AEZ model at all the lead times across all the AEZs. The $R^2$ scores were very high for forecasts in the very-arid zones, with HBM-AEZ having 0.97, 0.90, and 0.79 compared to 0.93, 0.86, and 0.75 for the BARDL-AEZ at 6, 8 and 10 weeks lead time, respectively. These scores indicate that the HBM was better at capturing the variability within the observed data than the BARDL model. In terms of the forecast errors (RMSE), the HBM-AEZ also

performed better than the BARDL-AEZ model, with lower RMSE scores across the lead times.





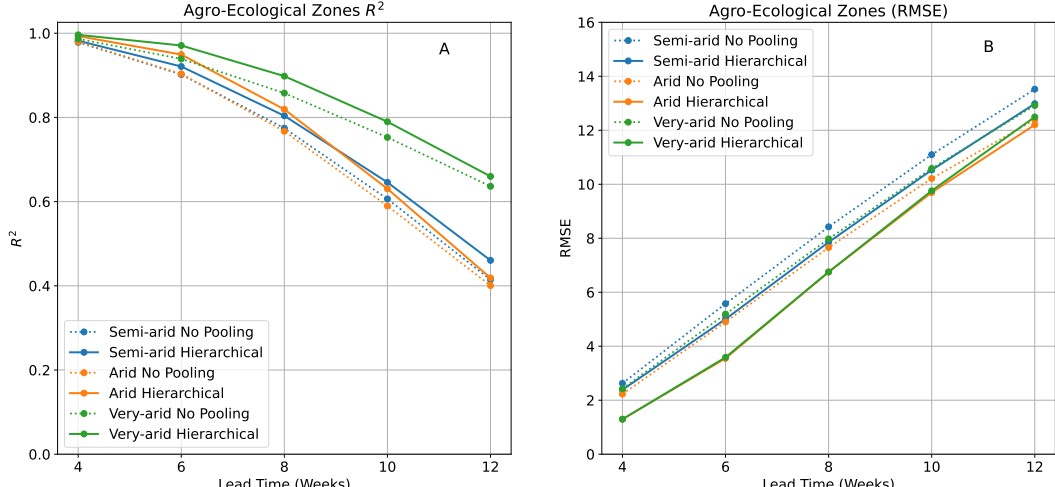

**Figure 6.** Plots showing $R^2$ Score (*left*) and RMSE (*right*) for BARDL-AEZ (Dotted) and HBM-AEZ (Solid) the VCI3M forecast over the different Agro-Ecological Zones

## 4.2 Model Performance for Land Cover Based Models

Figure 7 shows the performance metrics for the VCI3M forecast for the vegetation land covers. Overall, the HBM-LC performed better than the BARDL-LC except for the forest covers. (Where both models had almost identical $R^2$ scores across all lead times). The HBM-LC also performed well up to 10 weeks ahead for cropland with $R^2$ scores of 0.70 compared to 0.66 for
the BARDL model. The $R^2$ score for forecasts over shrublands and grasslands remained between 0.90 and 0.70 up to 8 weeks ahead for the HBM-LC. The forecast errors from the RMSE plot (figure 7 B), showed a slightly different pattern. The forecast errors for all the land covers except for forest covers were lower for the HBM-LC. There was, however, no difference in $R^2$ and RMSE over forest cover, probably because the group-level effects did not differ significantly from the global effects.


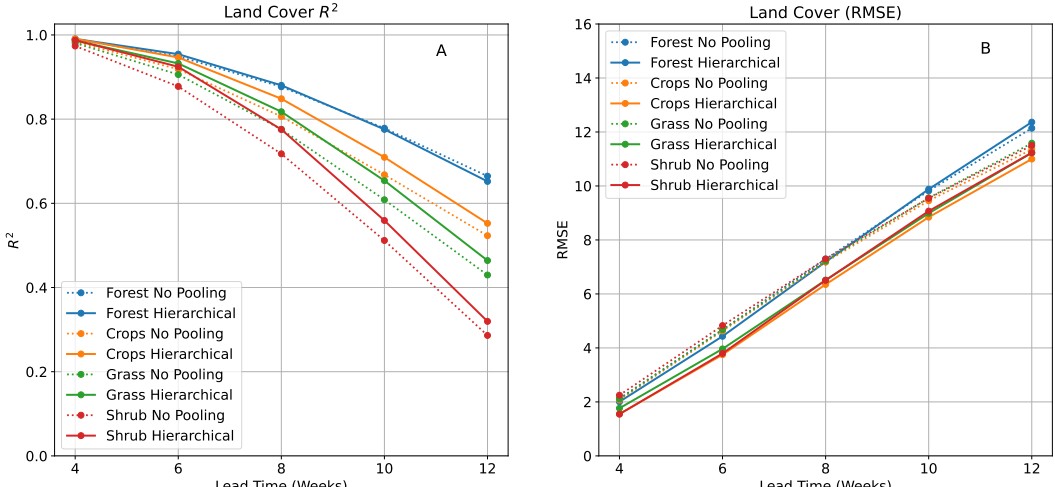

**Figure 7.** Plots showing R$^2$ (*left*) score and RMSE (*right*) for BARDL-LC (Dotted) and HBM-LC (Solid) the VCI3M forecast over the different vegetation land cover types.

### 4.3 Uncertainty Analysis

The forecast uncertainty of both forecasts models was analysed using the PICP and MPIW. The desired PICP value usually ranges between 0.90 to 0.99 Pang et al. (2018). If the PICP indicates, the number of times observed values fall within our forecast interval. On the other hand, the MPIW values show forecast precision and are expected to remain very low. Figure 8 shows the time series plots of forecast and observed VCI3M for the arid zone in Baringo county. Each plot shows the 95% prediction interval along with the PICP and MPIW for 4 to 10 weeks lead time. The PICP values for both models indicate

that observed values for all the lead times fall within a 95% credible interval of our forecast distributions over 90% of the time. The high PICP seen for the BARDL model from 8-Weeks was due to the wider forecast interval (error bars). A closer look at the MPIW values indicates that the HBM-AEZ forecasts are more precise than BARDL-AEZ, indicating that forecasts from HBM-AEZ have reduced uncertainties. A similar trend was seen for forecasts across all land covers. Overall, the MPIW metrics reiterate that forecasts by the HBM have lower errors than the BARDL. In addition, the partially pooled parameters

also mean errors from the HBM is a better representation of the actual forecast error. Thus, even though PICP from 10 weeks ahead seems high for the BARDL model, they do not reflect the truth.
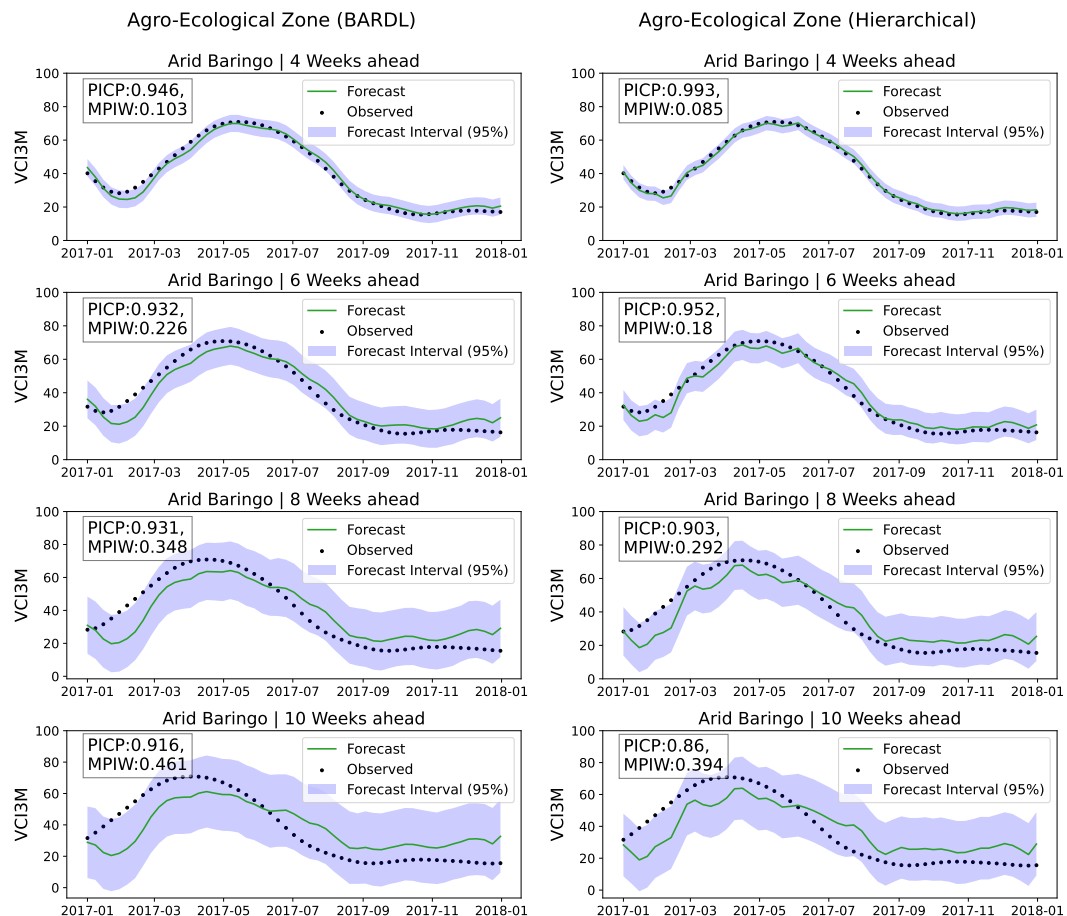

**Figure 8.** Plots showing forecast for Arid zones for 4 and 10 weeks lead times and their uncertainties (PICP & MPIW).

The mean PICP and MPIW for both AEZs and land covers over the selected counties are in tables B1 and B2.

## 4.4   Predicting Drought Event (ROC Curves)

Although our models produce accurate VCI3M values at the various lead times, our target users are also interested in whether
or not a drought event alarm will be triggered at a defined threshold. Therefore, the skill of the forecast models at predicting
drought events was assessed with the ROC curve with a threshold of VCI3M < 35%. VCI3M values below this threshold
depict moderate to severe drought conditions (Klisch and Atzberger, 2016).

The ROC plots in figure (9) shows the TPR (Hit rate) and FAR (False Alarm) for the three arid zones. The dot on the curves
indicates the VCI3M<35 threshold. Apart from the very-arid zones (fig 9 C), significant differences were seen between TPR
and FAR for drought events predicted by the HBM-AEZ compared to the BARDL-AEZ (fig 9 A & B) at all lead times. The
Hit rates for the HBM-AEZ were higher than the BARDL-AEZ and were mostly above 80% for drought events from 4 to 10
weeks ahead in the arid areas (fig 9 B) with false alarm rates between 1% to 18%. Drought events in semi-arid zones also had




hit rates above 80% up until 8 weeks (fig 9 B). Both models performed very well at detecting moderate to severe drought events in the very-arid zones, as seen in figure (9 C), which was mostly because of the frequent occurrence of drought events in the 305 very-arid zones.

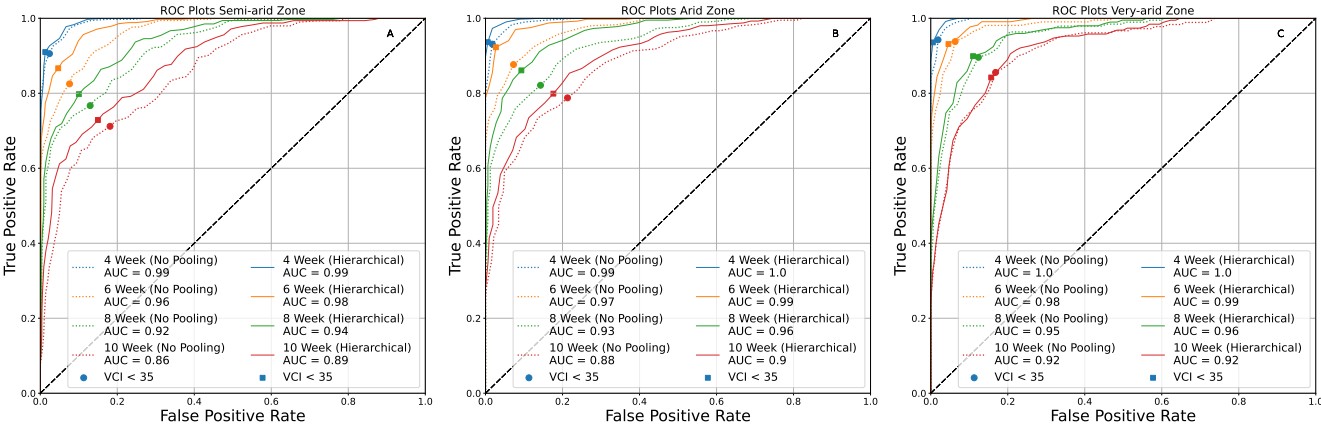

**Figure 9.** ROC plots generally showing higher Hit Rates for HBM in Semi Arid, Arid and Very Arid Zones

Figure 10 shows the ROC plots for the croplands, grasslands and shrubs for the BARDL-LC compared to HBM-LC. Overall, drought events predicted by the HBM-LC also had higher hit rates with lesser false alarm rates than the BARDL-LC model. The hit rates for drought events over croplands remained above 80% up to 10 weeks ahead, with false alarm rates ranging between 1% to 16%. The TPR for grasslands and shrubs dropped quickly after 6 weeks. The TPRs were generally above 60% 310 for all land covers at all lead times.

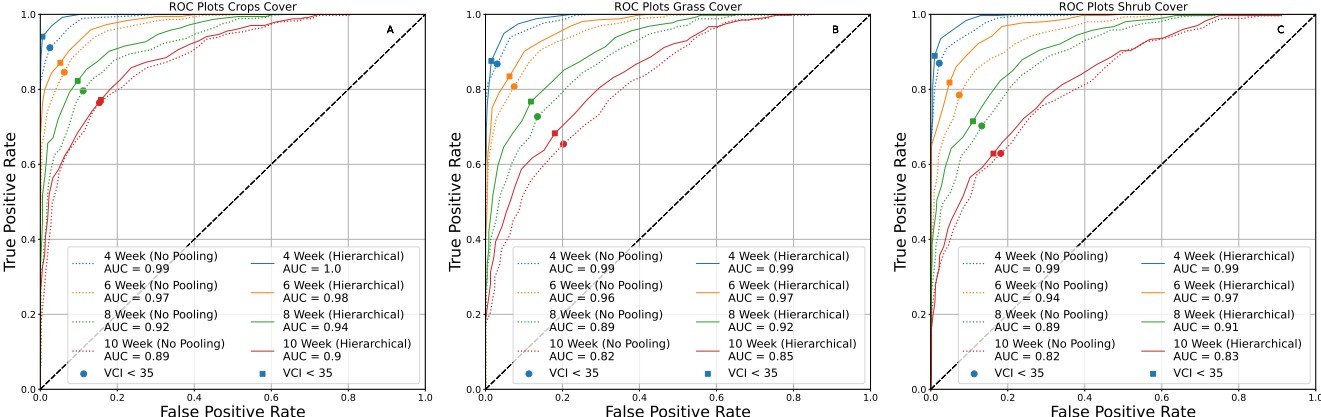

**Figure 10.** ROC Plots for Crops, Grass and Shrub Land Covers





## 4.5  Forecast Reliability

The reliability plots in figure 11 is a joint distribution of the binned forecast probabilities and relative frequency of the actual observed drought event (observed binaries = 1) for their respective probability bins. In a perfect system, the joint plots should lie on the diagonal line. The plots also show a histogram that depicts the model's sharpness. A perfect sharpness plot should

have peaks at the extreme ends of the histogram. A peak close to the 0% probability bin indicates the frequency at which the model predicted a 'No Drought' event. Whereas a peak close to the 100% probability bin means otherwise. It is essential to state that a forecast system is said to exhibit little or no sharpness when a sharpness peak is close to the long-term mean or climatology (Jolliffe and Stephenson, 2012).

The reliability diagrams for both BARDL-AEZ and HBM-AEZ (figure 11) showed some differences but were not very

significant. The proximity of the reliability curves to diagonal, especially for the arid zones (figure 11, plots (A & C)) indicates the forecast probabilities from both models can be trusted for early warning and early action. From plots (A), we see that when the BARDL-AEZ model predicts drought event with a probability ranging between 80% to 100% at 4 to 6 weeks ahead, the forecast probability agrees with the observed frequency 90% to 99% of the time, which can also be seen in plots (C) for the HBM-AEZ model. For the very-arid zones forecast probabilities between 60% to about 80% (figure 11, plot(B

& D)) corresponded with very high observed relative frequencies above 80%, a situation referred to as *'under forecasting'*. Under forecasting describes the situation where forecast probabilities do not adequately reflect observed events (Wilks, 2006). However, a closer look shows some subtle improvements with the HBM-AEZ, with a slight difference in the under forecasting effect from 4 to 6 week lead times. Regarding the sharpness of the models, a higher frequency of drought events was seen in the higher forecast probability bin for the HBM-AEZ 11, plot(C & D)) compared to the BARDL-AEZ 11, plot(A & B)) especially

from 6 to 12 weeks in the arid zone. The reliability diagrams for croplands and grasslands for both BARDL-LC and HBM-LC models also showed similar patterns. Please see figure C1 in Appendix C.

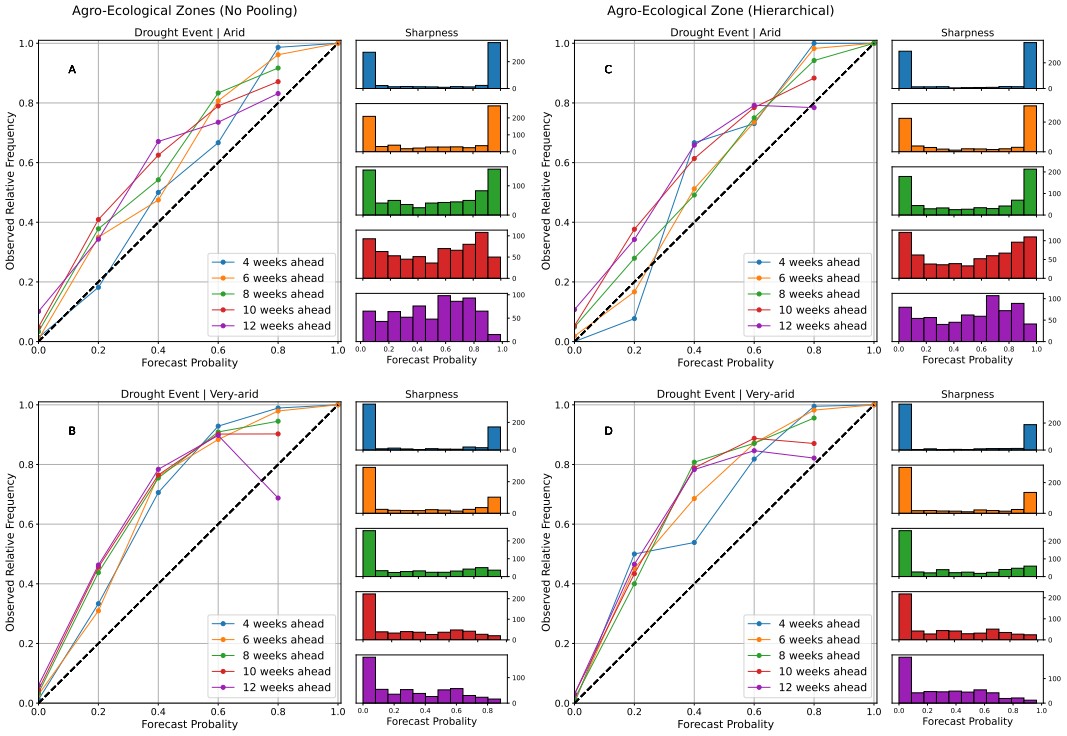

**Figure 11.** Reliability and sharpness plots showing a joint distribution of forecast probabilities and observed frequencies for various Arid and Very-Arid Agro-Ecological Zones for the different lead times

The skill of the models at predicting the onset and end of a drought period can be seen in figure 12. The figure shows a time series plot of observed and forecasted VCI3M at a 4-weeks lead time in a very-arid zone within Turkana county of Kenays for 2017. The plot also shows the forecast probability as a dot on vertical lines depicting the onset and end of a drought period. We can see from figure 12 (A) that at the start of a drought period where the observed VCI3M dropped below the threshold (VCI3M<35) line, the forecasted probability for the drought event predicted by the BARDL-AEZ was 9.4%. The low probability was because the forecasted VCI3M value model was higher than the observed value and threshold. However, the likelihood of a drought onset predicted by the HBM-AEZ in figure 12 (B) was 73%, prompting a trigger for early action. Towards the end of the drought period, the BARDL-AEZ model gave a high drought probability even though the drought duration had ended. Although these differences are not seen in all cases at the onset and end of a drought period, the few occurrences in some regions of interest emphasise that HBMs provide a better approach to forecasting VCI3M over a diverse region.

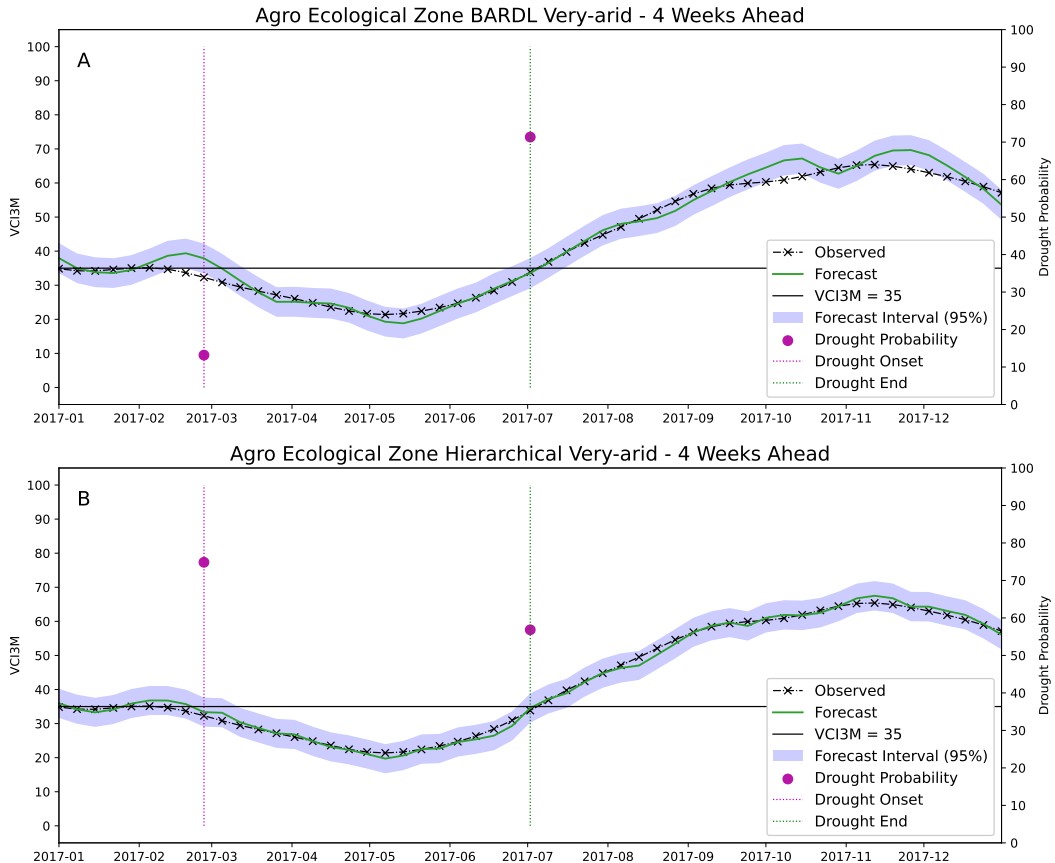

**Figure 12.** A time series plot showing the observed and forecasted VCI3M for the period of 2017. Forecast probabilities are indicated as points on the horizontal lines marking the onset and end of a drought periods

## 4.6 Test Transfer Learning

Although the data used for training and developing forecast models are usually sampled to represent a given area of interest, the

goal in most cases is to have models that can scale up to produce forecasts over more expansive areas. The second objective of this study was to test the transfer learning capability of HBMs over other regions. The partially pooled data used for hierarchical parameter approximations were sampled from 6 counties. The trained models for the different lead times were then used to forecast VCI3M for the AEZs, and land covers over ten additional counties (shown with black boundaries in figure2), which were not part of the training sets. The comparison of their $R^2$ and RMSE metrics in figure 13 proved that both HBMs were able

to forecast VCI3M over the non-trained counties accurately. For the AEZs, some significant differences were seen between the trained and non-trained counties in the semi-arid zones in terms of explained variances ($R^2$ score) (figure 13A). The case was different for forecast error in the same zone as seen in figure 13B. A significant gap was also seen for the forecast error over the very-arid area but not for the explained variances. Performance over the different land covers, however, remained very close,





especially for the RMSE (figure 13D) despite the gap seen for grassland in the $R^2$ score plots(figure 13C). These differences

can be linked to the fact that although some non-trained counties may have similar AEZs or land covers, their climatic and

vegetation phonology cycles are not similar. Aside from these observed differences, the HBMs could generalise and accurately

forecast when given new unseen data.

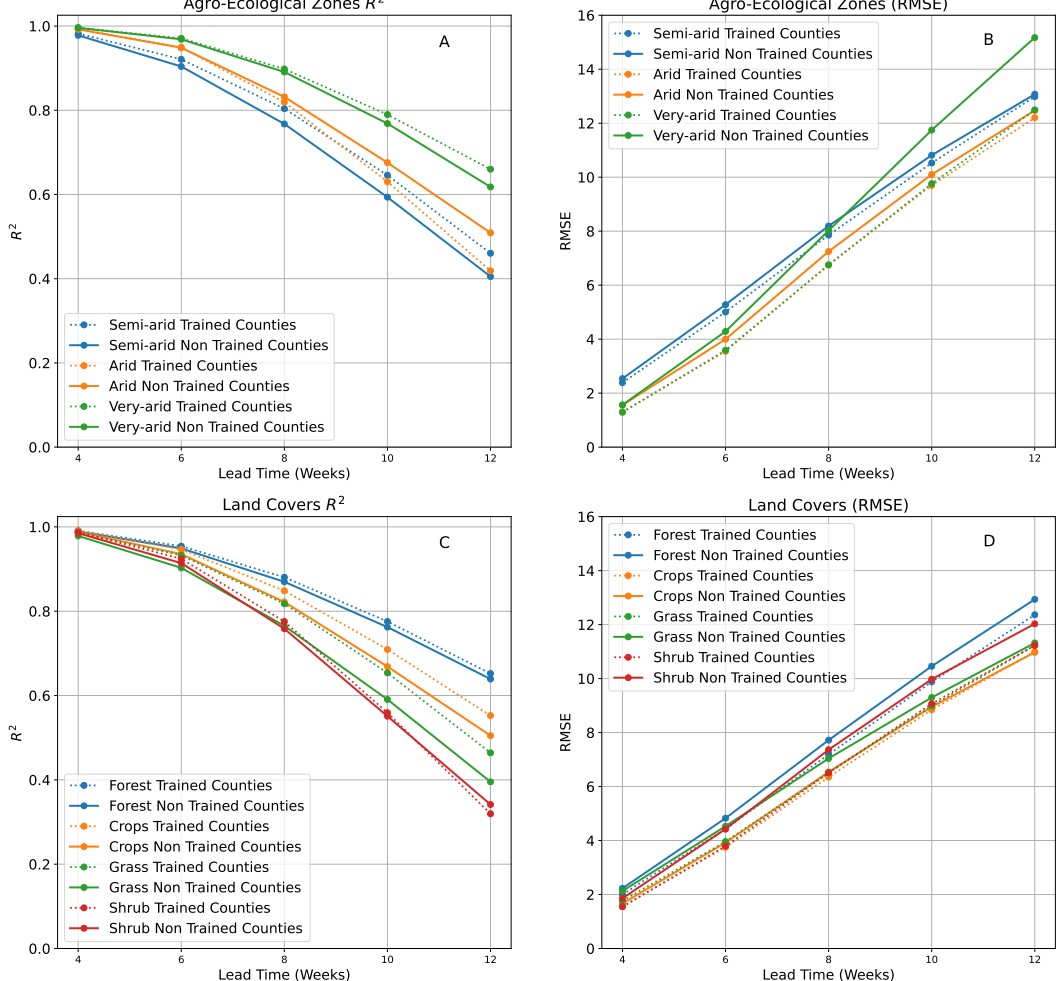

**Figure 13.** Plot showing $R^2$ score and RMSE for forecasts over counties not included in the training data used HBM (solid line) versus the counties included in the training data (dotted lines)

## 5    Discussion

In this paper, we sought to improve the forecast accuracy of VCI3M over vast areas with varying AEZs, and land covers using

an HBM. Compared to the non-hierarchical BARDL model, the HBM presented a more realistic approach for forecasting

VCI3M in regions with different AEZs or land covers. The evaluation of the HBM based on $R^2$ metrics indicated that forecasts



over the very-arid zones and forest cover areas showed higher accuracies at longer lead times. The high accuracy observed for the very-arid zones could be a consequence of the significant contribution from the lagged soil moisture to future VCI3M in addition to precipitation as seen in figures D1 and D2. For the forest areas, the observation could be because some dense forests show slight variation during drought periods.

The strong relationship between lagged soil moisture VCIM over forest areas could be due to the frequent precipitation and high soil moisture retention in areas. On the other hand, the low contribution of soil moisture to forecasts in croplands, grasslands, and shrubs could be attributed to the low soil moisture levels over grass and shrub areas (James et al., 2003; Tyagi et al., 2013). For croplands, the low contribution of soil moisture could be due to several factors, including high temperature and soil type. However, in the very-arid areas, the high relative importance of soil moisture could be due to the rapid response of vegetation to sudden increases in soil moisture, especially after long periods of dryness.

Overall, results from the various skill assessments showed that forecasts with HBM were more precise with a low probability of false alarms rate for drought events than the BARDL model. The HBM was also able to effectively identify drought events in counties with diverse AEZs and some land covers.

Relating the overall forecast skill assessments from this work to previous works, the HBM showed an approximately one week increase in the forecast range compared to the results from the BARDL method used in Salakpi et al. (2021). On average, the HBM also exhibited an approximately 2-weeks increase in forecast range, compared to the auto-regression method used in Salakpi et al. (2021) and Barrett et al. (2020). Furthermore, using the HBM also enabled the simultaneous forecast of VCI3M for different AEZs and land covers which we could not do with the methods used in (Salakpi et al., 2021) and (Barrett et al., 2020). Finally, despite the improvement seen with the HBM, the BARDL models also proved to be useful at predicting drought events at the set threshold as demonstrated by (Salakpi et al., 2021).

Aside from the improvement in the forecast range, the HBM also had some added strengths. First of all, the hierarchical nature of the model parameters (see figure 3) enabled the incorporation of the varying (AEZs or land covers) effects of climate and other biophysical factors on vegetation conditions. Thus, modelling within the HBM framework made it possible to learn the within-sample parameters in addition to the global parameters and accurately forecast VCI3M values specific to the AEZs and land covers. Secondly, modelling within a Bayesian context means the model outputs probability distributions instead of point values. These distributions present a direct approach to quantifying forecast uncertainties. The probability distribution of forecasts also made it possible to derive forecast probabilities, which allowed us to quantify the likelihood of drought events in different locations. Finally, the HBM also makes it possible to transfer trained models to similar datasets that were not part of the initial training data. Transferring the model also means even though the HBM model was calibrated on the data from Kenya, it can be scaled up to generate forecasts for wider regions without the need to re-calibrate.

The threat of agricultural drought to food security and global economies has pushed agencies like the USAID and FAO to develop early warning systems that continually monitor drought events. However, agricultural drought over vast and diverse ASAL regions poses a challenge to effective monitoring Boken et al. (2005); Vicente-Serrano (2006). Policy and decision-makers at these agencies, including Kenya's National Drought Management Authority (NDMA), our primary stakeholder, can incorporate the HBM demonstrated in this paper into their existing early warning systems to enhance their efforts. Aside from



accounting for the different AEZs or land covers, the forecasted drought probabilities from the HBM will also enable intelligent decision making for drought relief agencies that practice the Forecast based Financing (FbF) (Coughlan de Perez et al., 2015) for drought early action.

The methods used in this paper also had a few limitations. A fundamental limitation was the timely availability of the ESA CCI Soil Moisture data. A setback that can affect the prospects of producing real-time forecasts. Parameter inference via HMC sampler also takes a long time to complete partly due to the complex nature of the HBM and the number of data points involved. However, this was not considered a significant limitation as it only occurs during the model training phase. Once the model converges, and sampling completes, the posterior predictive sampling or forecasting VCI3M takes seconds.

**6   Conclusion and Future Work**

In this paper, we presented a proof-of-concept that HBM can factor spatial differences into drought forecast. Using this approach also allowed us to understand the vegetation dynamics in Agro-climatic areas and regions with diverse vegetation covers. For instance, we saw an approximately one week gain in forecast range for vegetation conditions in very-arid as well as forests (Tree cover) and cropping areas. Furthermore, we have shown that soil moisture contributes more when forecasting
VCI3M over very-arid areas and forest covers. We also showed that HBM trained with data in one area could be transferred to other similar datasets in other regions. Future research work should consider more complex HBMs that takes into account variations for different land cover types within the various Agro-Ecological zones and the seasonal differences.




*Code and data availability.* Link to Data and Code repository https://github.com/edd3x/Hierarchical-Bayesian-ARDL.git

## Appendix A: Forecast Metrics Semi-Humid and Humid Zones

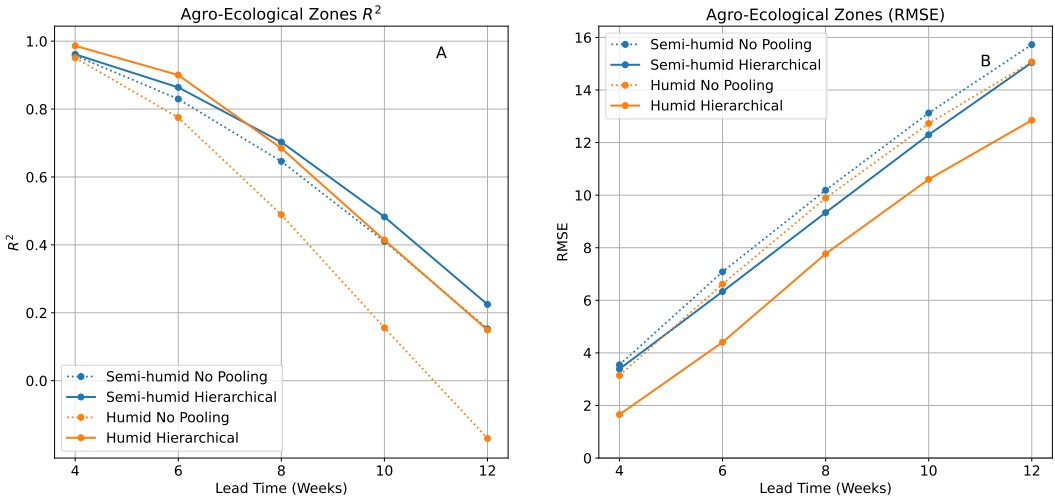

**Figure A1.** Plots showing $R^2$ Score and RMSE for BARDL-AEZ (Dotted) and HBM-AEZ (Solid) the VCI3M forecast over the different humid zones





**Appendix B: PICP and MPIW for Land Covers and Agro-Ecological Zones**

**Table B1.** Table showing a PICP and MPIW (In Parenthesis) for the various Agro-Ecological Zones

| Models | AEZ | 4 | 6 | 8 | 10 | 12 |
|---|---|---|---|---|---|---|
| BARDL | Humid | 0.88 (0.09) | 0.88 (0.21) | 0.9 (0.32) | 0.91 (0.42) | 0.91 (0.52) |
| | Semi-Humid | 0.87 (0.1) | 0.88 (0.22) | 0.88 (0.33) | 0.89 (0.43) | 0.9 (0.52) |
| | Semi-Arid | 0.97 (0.1) | 0.96 (0.22) | 0.94 (0.32) | 0.95 (0.42) | 0.95 (0.5) |
| | Arid | 0.98 (0.11) | 0.98 (0.23) | 0.97 (0.33) | 0.97 (0.43) | 0.96 (0.51) |
| | Very-Arid | 0.96 (0.11) | 0.95 (0.23) | 0.94 (0.33) | 0.94 (0.42) | 0.94 (0.5) |
| Hierarchical | Humid | 0.97 (0.09) | 0.95 (0.18) | 0.94 (0.29) | 0.94 (0.4) | 0.93 (0.48) |
| | Semi-Humid | 0.81 (0.09) | 0.84 (0.18) | 0.88 (0.29) | 0.88 (0.39) | 0.88 (0.48) |
| | Semi-Arid | 0.94 (0.09) | 0.94 (0.18) | 0.95 (0.29) | 0.95 (0.39) | 0.95 (0.48) |
| | Arid | 1.0 (0.09) | 0.98 (0.18) | 0.96 (0.29) | 0.95 (0.39) | 0.94 (0.48) |
| | Very-Arid | 1.0 (0.09) | 0.97 (0.18) | 0.94 (0.29) | 0.93 (0.39) | 0.93 (0.48) |

**Table B2.** Table showing a PICP and MPIW (In Parenthesis) for the various vegetation land covers

| Model | Land Covers | 4 | 6 | 8 | 10 | 12 |
|---|---|---|---|---|---|---|
| BARDL | Forest | 0.97 (0.09) | 0.96 (0.19) | 0.95 (0.29) | 0.95 (0.38) | 0.94 (0.46) |
| | Crops | 0.97 (0.09) | 0.95 (0.19) | 0.94 (0.29) | 0.95 (0.38) | 0.95 (0.46) |
| | Grass | 0.97 (0.09) | 0.96 (0.19) | 0.96 (0.29) | 0.97 (0.38) | 0.96 (0.46) |
| | Shrub | 0.96 (0.09) | 0.96 (0.19) | 0.96 (0.29) | 0.96 (0.38) | 0.96 (0.46) |
| Hierarchical | Forest | 0.94 (0.08) | 0.93 (0.17) | 0.94 (0.27) | 0.94 (0.37) | 0.94 (0.46) |
| | Crops | 0.99 (0.08) | 0.97 (0.17) | 0.96 (0.27) | 0.95 (0.37) | 0.95 (0.46) |
| | Grass | 0.98 (0.08) | 0.97 (0.17) | 0.96 (0.27) | 0.96 (0.37) | 0.95 (0.46) |
| | Shrub | 0.98 (0.08) | 0.98 (0.17) | 0.98 (0.27) | 0.97 (0.37) | 0.96 (0.46) |





# Appendix C: Reliability Diagram for Crop and Grass Covers

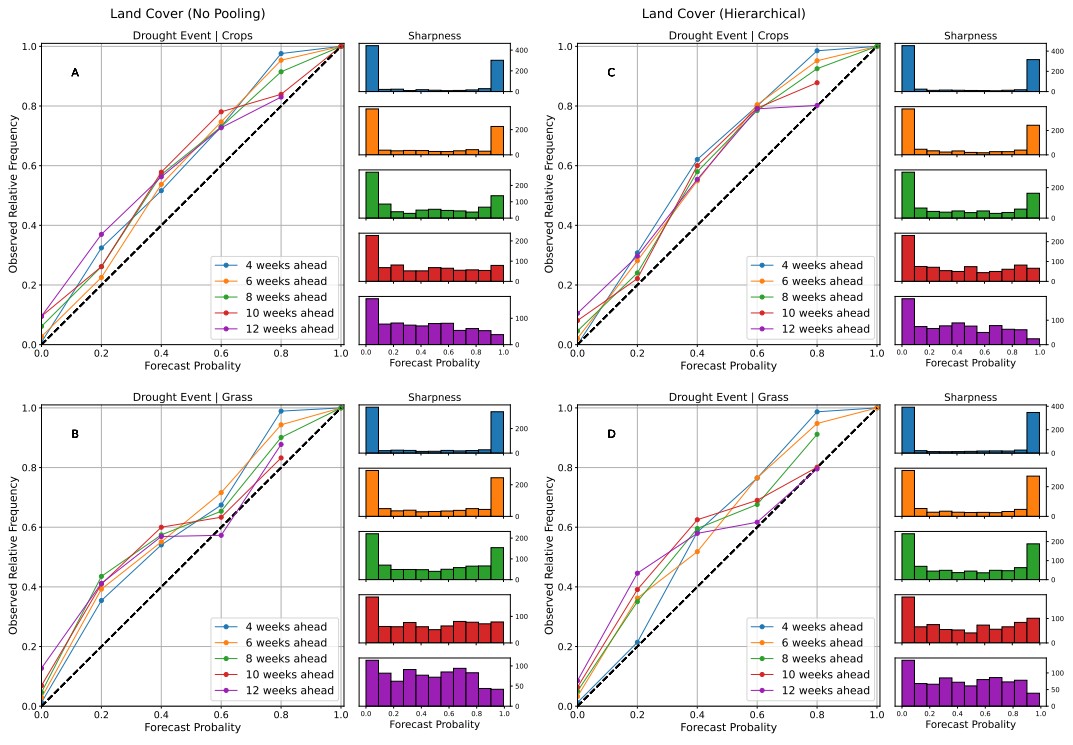

**Figure C1.** Reliability and sharpness plots showing a joint distribution of forecast probabilities and observed frequencies for various Agro-Ecological Zones and Land Cover for different lead times



## Appendix D: Percentage Relative Importance

Figure D2. Plots showing the relative importance of the lagged input variables (VCI3M, P3M, SM3M) and VCI3M at 4 to 12 lead times the different vegetation land covers





*Author contributions.* EES lead author, data preprocessing, modelling (Coding) & running BARDL and HBM methods; JMM data acquisition, preprocessing, cartography and feedback; AB code for smoothing time series data; SO, PR, & PH conceptualised the initial idea and
provided supervision and feedback; The final manuscript was edited and reviewed by all authors.

*Competing interests.* All authors of the paper declare no known competing interests (financial, personal relationships) that could have influenced this study.

*Acknowledgements.* The work was funded by the UK Newton Fund's Development in Africa with Radio Astronomy (DARA) Big Data
project delivered via STFC with grant number ST/R001898/1 and by the Science for Humanitarian Emergencies and Resilience (SHEAR)
consortium project 'Towards Forecast-based Preparedness Action' (ForPAc, www.forpac.org), Grant Number NE/P000673/1, funded by
the UK Natural Environment Research Council (NERC), the Economic and Social Research Council (ESRC), and the UK Department for
International Development (DfID).





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




## List of Figures





**List of Tables**

570

**Figure D1.** Plots showing the relative importance of the lagged input variables (VCI3M, P3M, SM3M) and VCI3M at 4 to 12 lead times the different Agro-Ecological zones