# Peer review of "A Dynamic Hierarchical Bayesian Approach for Forecasting Vegetation Condition"

_Natural Hazards and Earth System Sciences, 2021_

## Author Response (AR1)

**A Dynamic Hierarchical Bayesian Approach for Forecasting Vegetation Condition**

**Color Code:**
- **Comments (Gray)**
- **Response (Blue)**

Reviewer 1

Review of: A Dynamic Hierarchical Bayesian Approach for Forecasting Vegetation Condition. This study proposed a method to forecast vegetation condition index by means of soil moisture and precipitation data in Kenya by means of Bayesian method. I find the study not very well structured and difficult to follow given lack of information on the data sources used and period of analysis. In any case the justification of the study is mostly related to the assessment of drought forecasting but I cannot find any analysis and result that can be directly focused on drought analysis. The authors should detail better and illustrate more if long-time series are used and if forecasting is useful during critical drought periods. See specific comments below. They refer to specific lines of the manuscript:

Response:

This paper is part 2 of a previous paper (https://doi.org/10.5194/nhess-2021-223) and uses the same data (reference was made to this in section 2.1.). But I realise this was the wrong decision, so a full description of the data used for this research will be added to the data section.

Secondly, the entire premise of this paper was based on our earlier paper, Barrett et al. 2020 (https://www.sciencedirect.com/science/article/pii/S003442572030256X). We aimed to improve the precision and accuracy of the VCI forecast beyond the six weeks achieved in the  Barrett et al. 2020. Thus, the analysis of our results mainly focused on the accuracy and precision of the VCI forecast beyond six weeks. And more specifically, how our proposed model differentiated VCI forecast within a given region of interest.

The research was in partnership with the NDMA. They are currently using VCI for monitoring drought. They have used the indicator extensively for their monthly drought reports and bulletins.

However, all the issues raised about the paper structure are well noted and will be addressed accordingly.

16. Does this number refer to the global scale? What is the source?

    Response:

    Yes, this is on a global scale Source:
    https://wwfint.awsassets.panda.org/downloads/drought_risk__wwf_.pdf . The source of the information was not properly cited, and this has been fixed in line 16/ page 1

17. Note that enhanced atmospheric evaporative demand increases the severity of agricultural droughts, particularly under low soil moisture conditions.

    Response:

    We agree that atmospheric evaporative demand is vital for agricultural drought studies. Information on atmospheric evaporative demand as a recommendation for future work in line 434/page26.

18. But also human practices may reinforce drought severity or reduce it as a function of soil management, and crop rotation. See the several studies by Prof. Rattan Lal about this issue.

    Response:

    Comment accepted and well noted; information on this will also be included in the paper's literature review section lines 22-24/page2.

19. I would say that ecological droughts and hydrological droughts are also strongly complex. I would qualify this issue.

    Response:

    This is also well noted the sentence in lines 22-24/page2 has been amended to reflect this.

24-24. I would include issues related to crop practices, crop types, etc.

    Response:

    Comment well noted and accepted; issues on crop practices have been included in line 27/page 2.

    36-88. In the introduction, I am surprised that dynamic forecasting of drought based on climate models is not considered (e.g. https://www.nature.com/articles/s41612-021-00189-4). There is a large research topic on this issue that should be mentioned/discussed in relation to the statistical techniques proposed in this article.

Response:

Comment well noted; however, the focus of this paper was using the Hierarchical Bayesian Model to provide differentiated forecasts within a given region of interest and I do not see how this links to the suggested paper.

125.     What are the sources of precipitation, VCI and soil moisture? This must be detailed and discussed in relation to the availability of the data, quality, time period of analysis, etc. There is not sufficient information to determine which data the authors are using.

Response:

We accept that it was a wrong decision to refer the reader to data sources in a previous paper (https://doi.org/10.5194/nhess-2021-223). This has been corrected from lines 105-118/page6.

126.     What are the variables that are intended to be forecasted (precipitation? Soil moisture? VCI? All of them?) This is not clear in the methodology what is the target variable and what is the usefulness of the other variables. Maybe are precipitation and soil moisture possible drivers of the VCI and they are used to generate a predictive model of VCI? This should be clarified. This is indicated below 180-220; but it should be explained in detail before to avoid confusion. I would also suggest atmospheric evaporative demand as predictor of the VCI as several studies have demonstrated strong importance on agricultural and ecological drought conditions.

Response:

The target variable is the VCI and lags of precipitation and soil moisture are being used as drivers for the VCI in this context. The comment is well noted and will be clarified as suggested.

In this paper, we considered variables that could be directly derived from satellite remote sensing products. However, the suggestion of the atmospheric evaporative demand ihas been to the recommendation for future work in line 434/page26

Figure 5. Why is only one year of results showed in this plot? I would suggest to include longer time series as we cannot determine if this year correspond to drought or to normal and humid conditions.

Response:

Unfortunately, data was limited by access to readily available soil moisture data. At the time of data acquisition and preprocessing our soil moisture data was only up to 2018. HBM was

calibrated to data from 2001 to 2016 and evaluated on 2017 and 2018 data which is why the model evaluation was a single year.

Figure 6. What about seasonality? Is the model capable to forecast vegetation conditions with the same accuracy for seasons of high or low vegetation activity? This should be at least discussed. What is the different performance between drought and non-drought years?

Response:

This comment is also well noted, analysis of the seasonality section has been added under results from lines 293/page 15 and line 395/page25

Figure 7. I see not only agricultural areas are considered in the study but also forest lands so introduction should be modified to also focus on ecological droughts.

Basically all the figures are showing a single year for the VCI forecasting and it is not clear for me if this is the period of analysis or it is an arbitrary selection. In any case. This must be clarified as I do not think robust results and conclusions related to the suggested methodological approach can be obtained from the application to a single year.

Response:

From literature, the definition of ecological drought falls under agriculture drought thus including a definition will be duplication. We used a single year because the aim here was to show how well the HBM forecasted VCI for different agro-ecological zones and vegetation land covers for up to 12 weeks ahead.

366.    Where is this analysed? I cannot find a plot in which this relationship is showed.

Response:

These can be seen in the Percentage Relative Importance plots figures E1 and E2. This has been fixed by referencing the figures in the discussion section from lines 387-393/page25.

367.    A reason to include also atmospheric evaporative demand as predictor...

Response:

Well noted, this has been added to future work line 434/page26. It will be considered in future work as the funding phase for this project has ended.

Reviewer 2

General comments

The paper has objectives similar to those of the paper by Salakpi et al., NHESSD, "Forecasting vegetation conditions with a Bayesian auto-regressive distributed lags (BARDL) model (https://doi.org/10.5194/nhess-2021-223 ). The aim is to forecast vegetation conditions leading to agricultural droughts. The paper presents a new method, as compared to the previous paper, based on a dynamic Hierarchical Bayesian Model (HBM). The data used in the two papers are the same, and the method for assessing the forecasts are also similar. The HBM model has the advantage of providing differentiated forecasts within the same region, as compared to BARDL that provides homogeneous forecasts within each region. The BARDL model is used as benchmark for assessing the improvements provided by the HBM model. There are therefore lots of repetitions between the two papers. The present one refers to the previous one to present the data used in the study. This is not acceptable, as a paper should be self-contained and a minimum of information should be provided about data. The problem would be solved if the two papers were merged in a unique paper.

Concerning the present paper, I find it is well written and the analysis is sounded and honest. I only have a few minor comments listed below.

Response:

Thank you for the feedback, the issues on the omission of the data description are well noted and will be fixed in the final draft of the paper. The issue with repetitions is also noted and instead of merging, both papers will be published as parts A and B.

Specific comments

1/ P.1 line 16: the number of affected persons is different than in the previous paper

Response:

Comment well noted, this was an error and has been corrected in that paper, the correct figure is the one stated in this paper.

2/P.5. Section 2.1 should be developed

Response:

Comment noted and accepted, details on the data description has been added lines 105-118/page6.

3/ p. 6 line 116: ROI is not defined

Response:

Comment well noted and will be fixed. This was an oversight and has been addressed ROI changed to region in line 135/page7

4/ p.7 line 135: the authors mention that seasonal means are subtracted, but how are the seasons defined?

Response:

That statement should read annual mean per land cover type and AEZ is subtracted. The error corrected from line 154-156/page7

5/ p.9 lines 172-175: this section is not clear

Response:

Comment well-noted sentence should not have started with "However" that has been fixed and sentences in that section have been restructured to make it clear to the reader line 192-193/page 10.

6/ Figure 4: I do not understand how to read this figure

Response:

The figure shows how the HBM is structured. The figure depicts how the model parameters and data are defined and how it all fits together. The caption of the figure 4/page 11has been updated.

7/ p.10 eq.(3). This is strange to present the equation of the model after the presentation of the HBM model that refers to the parameters listed in Eq. (3).

Response:

Comment well noted and this will be rearranged in a proper order that makes more sense lines 202-214/pages 10-11

8/ p.12 line 253. The BARDL model based on a "no-pooling" is the same as the model presented in the previous paper, isn't it? If this is true, this should be stated.

Response:

Yes it's the same approach used in the previous paper, this was mentioned from line 56/page 3

9/ p.12 line 261: the differences between the two models is Figure 5 are not so large. So the claim that HBM model is more accurate than BARDL is not fully justified.

Response:

The statement will be toned down, rather the emphasis will be on the improvement in the lead times. Figure 5 and the narrative has in been dropped from the paper.

10/ Revise Figures 6, 7, 9, 10, 13 as the full line and dotted lines cannot be distinguished.

Response:

Comment well-noted figures have been enlarged to make them more legible to readers.

11/ p. 20 section test transfer learning. From Figure 2, it is not easy to see which counties are used for calibration and for validation.

Response:

Comment well-noted, the make has been made larger to make the black boundary lines indicating the Non-Train counties in the figure more visible.

---

## Author Response (AR2)

Response to Comments:
Paper Title: "A Dynamic Hierarchical Bayesian Approach for Forecasting Vegetation Condition"

Comment 1
In the List of Figures (Pag. 38), Figure E1 should be before Figure E2.
Figure E1 is not included in Appendix E: Percentage Relative Importance

Response:
The document has been formatted to correct the position of Figures E1 and E2. The list of figures has been updated to reflect the changes. Figure E2 is now Figure F1: Percentage Relative Importance (Land Cover) and Figure E1 is the same with updated title Percentage Relative Importance (Agro-Ecological zones)

Comment 2
Please ensure that the colour schemes used in your maps and charts allow readers with colour vision deficiencies to correctly interpret your findings. Please check your figures using the Coblis – Color Blindness Simulator (https://www.color-blindness.com/coblis-color-blindness-simulator/) and revise the colour schemes accordingly.

Response:
All the figures were checked with the Colour Blindness Simulator. Most of them passed except for the $R^2$, RMSE Forecast Probability, Relative Importance and Reliability Plots. These have been redone with colour schemes suitable colour deficient readers.